# Chitosan/Albumin Coating Factorial Optimization of Alginate/Dextran Sulfate Cores for Oral Delivery of Insulin

**DOI:** 10.3390/md21030179

**Published:** 2023-03-14

**Authors:** Bruno Pessoa, Mar Collado-Gonzalez, Giuseppina Sandri, António Ribeiro

**Affiliations:** 1Faculty of Pharmacy, University of Coimbra, 3000-048 Coimbra, Portugal; 2Department of Cell Biology and Histology, Faculty of Biology, University of Murcia, 30100 Murcia, Spain; 3Department of Drug Sciences, University of Pavia, Viale Taramelli, 12, 27100 Pavia, Italy; 4i_3_S, IBMC, Rua Alfredo Allen, 4200-135 Porto, Portugal

**Keywords:** biopolymers, Box–Behnken, factorial optimization, insulin delivery, ionotropic gelation, nanoparticles, polyelectrolyte complexation

## Abstract

The design of nanoparticle formulations composed of biopolymers, that govern the physicochemical properties of orally delivered insulin, relies on improving insulin stability and absorption through the intestinal mucosa while protecting it from harsh conditions in the gastrointestinal (GI) tract. Chitosan/polyethylene glycol (PEG) and albumin coating of alginate/dextran sulfate hydrogel cores are presented as a multilayer complex protecting insulin within the nanoparticle. This study aims to optimize a nanoparticle formulation by assessing the relationship between design parameters and experimental data using response surface methodology through a 3-factor 3-level optimization Box–Behnken design. While the selected independent variables were the concentrations of PEG, chitosan and albumin, the dependent variables were particle size, polydispersity index (PDI), zeta potential, and insulin release. Experimental results showed a nanoparticle size ranging from 313 to 585 nm, with PDI from 0.17 to 0.39 and zeta potential ranging from −29 to −44 mV. Insulin bioactivity was maintained in simulated GI media with over 45% cumulative release after 180 min in a simulated intestinal medium. Based on the experimental responses and according to the criteria of desirability on the experimental region’s constraints, solutions of 0.03% PEG, 0.047% chitosan and 1.20% albumin provide an optimum nanoparticle formulation for insulin oral delivery.

## 1. Introduction

The feasibility of innovative insulin delivery systems, which is viewed as a valid alternative to enable reduction of the number of injections [1], has been investigated in clinical studies, and more recently, the results of a phase 3a study revealed the superiority of a weekly insulin injection (insulin icodec) in decreasing HbA1c when compared with once daily insulin glargine in people with type 2 diabetes [2]. An oral long-acting acylated insulin analogue co-formulated with an absorption enhancer (I338) assessed in an 8-week trial in people with type 2 diabetes treated with oral glucose-lowering drugs, showed no difference in the magnitude of hypoglycemia or rates of adverse events detected in people randomized to I338 vs. insulin glargine [3]. The possibility of developing oral insulin formulations for the treatment of type 1 and type 2 diabetes continues to be explored [4]. It is currently in phase 3 studies, but according to the press release on the clinical trial sponsor’s website, the endpoints were not met [5].

Nanoparticles have been used widely for the oral delivery of biopharmaceuticals such as insulin. Nanoparticles stabilize active biomolecules of interest against harsh gastrointestinal (GI) conditions and ensure biological activity during manufacturing processes and transit through the GI tract [6]. Developing and optimizing nanoparticle-based formulations based on physicochemical and physiological parameters involves several and often connected processes [7]. Oral insulin delivery relies on nanoparticle properties. Therefore, nanoparticles, including biopolymers that show favorable characteristics for insulin retention, protection, absorption across the GI tract and targeted delivery, are a promising approach [6,8]. The ideal techniques to develop nanoparticles for protein and peptide drugs should avoid using solvents and harsh chemical conditions to ensure insulin activity after manufacturing [9,10].

Biopolymer blend-based nanoparticles have been used with success for oral delivery of insulin. Examples include alginate and chitosan [11], sterculia gum and chitosan [12], and dextran sulfate and chitosan [13]. Among their advantages are biocompatibility, biodegradability and multipurpose functions assigned to biopolymers [14].

Nanoparticle formulations prepared by ionotropic pregelation and containing insulin consist of an internal multilayer complex where insulin is protected within the nanoparticle and an outer coat consisting of protein with protease protection properties. The internal particle core consists of the polysaccharides alginate and dextran sulfate. Insulin is retained thanks to complexation with polysaccharide chitosan and further coating with bovine serum albumin [6].

Alginate consists of anionic polymeric chains of mannuronic acid and guluronic acid with biodegradable and biocompatible properties. It forms stable hydrogels in the presence of multivalent cations such as calcium and zinc due to intramolecular and intermolecular crosslinking of polymer chains [15,16]. Chitosan consists of unbranched polymer glucosamine and N-acetyl glucosamine chains. Additionally, chitosan shows biodegradable and biocompatible properties. Chitosan stabilizes alginate-based hydrogels during the formulation of nanoparticles and enhances insulin absorption through the paracellular route [17]. Chitosan coating is essential to strengthen the alginate/dextran sulfate core [18] and to incorporate other polyanionic biopolymers that increase insulin stability against GI enzymes and pH and to modulate insulin release in the GI tract [19,20].

Dextran sulfate is a branched polysaccharide consisting of α-1–6 linked glucose residues in the main chain and α-1–3 linked glucose in ramifications. This polysaccharide has 2.3 negative charges per monomer [21] and thus interacts easily with polycations such as chitosan. Stabilizers such as polyethylene glycol (PEG) have been shown to maintain the structural properties of nanoparticle-based formulations [22] during manufacturing or storage by providing stability in aqueous suspension and impacting the interaction between particles and other biological environmental components, including enzymes, cells and membranes. Albumin coating has been shown to minimize insulin degradation under GI conditions. This is because albumin could act as a protein sacrificial target for local enzymatic degradation [23,24]. Premature insulin release and degradation stand out among the factors that counteract the pharmacological action of orally delivered insulin [13].

We have optimized the properties of multilayer complex nanoparticles for specific purposes, such as nanoparticle size and stability. Among the several components, chitosan has significantly impacted nanoparticle physicochemical properties [13], which is especially interesting given that chitosan has been one of the most challenging components to standardize in biopolymer-based nanoparticle formulations [25]. The amount of chitosan used in biopolymer-based nanoparticles has been shown to play a principal role in drug release from chitosan complexes with alginate [15] and zein [26,27]. On the other hand, as nanotechnology characterization tools have been improved in recent years, major concerns have been raised regarding the stability of nanoparticles, thus justifying a complete and comprehensive characterization of nanoparticles during experiments [28]. Chitosan stabilizes polyanionic nanoparticles due to its polycationic nature [29] and its conformation in solution [30].

Suspensions of biopolymer-based nanoparticles remain stable because of the electrostatic repulsion forces between negative charges on their surface [31]. Furthermore, a prominent role played by the steric hindrance of polymers such as PEG [32,33] has been described in biopolymer-based formulations prepared by ionotropic gelation.

As nanoparticle formulations comprising biopolymers, such as chitosan and albumin, are associated with a high number of process and formulation factors, multiparametric design can be an excellent approach to assess the impact of several factors on nanoparticle features related to physicochemical properties and biological performance [34]. Experimental design has been applied to optimize nanoparticle-based formulations considering several advantages outcomes, including a reduction in the number of experimental runs, development of models to evaluate the relevance and statistical significance of studied factor effects, and evaluation of eventual interaction effect between factors [35,36]. Box–Behnken designs are 3-level factorial designs that have shown success as tools for optimization of formulations following response surface methodology because they permit assessment of the parameters, design of sequential designs and detection of eventual lack of model fit [37]. In the present study, a design of 15 experimental runs is proposed, for which a quadratic model is generated as follows:(1)$$Y=b_{0}+b_{1}X_{1}+b_{2}X_{2}+b_{3}X_{3}+b_{12}X_{1}X_{2}+b_{13}X_{1}X_{3}+b_{23}X_{2}X_{3}+b_{11}X_{1}^{2}+b_{22}X_{2}^{2}+b_{33}X_{3}^{2}$$
where Y is the measured dependent variable related to each factor level combination, b_0_ is an intercept, b_1_ to b_33_ are regression coefficients computed from experimental runs, and X_1_, X_2_ and X_3_ are the coded levels of independent variables. The terms X_i_, $X_{i}^{2}$ and X_i_X_j_ (i and j = 1, 2 or 3) correspond to a linear effect, quadratic effect and interactions, respectively [38].

Chitosan/albumin coating of alginate/dextran sulfate cores led to adequate nanoparticles for the oral delivery of insulin [39]. However, the reduction in the previously standardized chitosan amount of nanoparticles showed opposite effects on the size and stability of the nanoparticles [13].

The novelty of this study lies in the optimization of the chitosan/albumin coating step of alginate/dextran sulfate cores for oral delivery of insulin and the investigation of the eventual relationship between studied design factors and obtained experimental responses through the combination of response surface methodology with Box–Behnken design. The experimental design is based on previous knowledge related to the effect of PEG, chitosan and albumin on the nanoparticle structure and release properties, not neglecting the monitoring of insulin bioactivity during nanoparticle development and release studies. The optimum nanoparticle formulation is developed based on the effect of PEG, chitosan and albumin on minimizing nanoparticle size to increase particle uptake, minimizing granulometric size distribution to predict enhanced drug transport, reducing zeta potential to values lower than −30 mV to increase particle stability in suspension, minimizing insulin release in gastric conditions for insulin protection against gastric pH and enzymes, and maximizing insulin release in simulated intestinal conditions to improve insulin absorption across the GI epithelium.

## 2. Results and Discussion

### 2.1. Preparation and Characterization of Nanoparticles

The Box–Behnken design was performed to assess any relationship between formulation components PEG, chitosan and albumin on physicochemical properties of nanoparticles for the optimization of formulation for oral insulin delivery. Insulin nanoparticles were prepared by ionotropic gelation followed by complexation [13,40]. The selection of dependent variables is aimed at a comprehensive characterization of nanoparticles, including size, PDI, zeta potential and insulin release behavior, which are considered critical for improving the oral bioavailability of proteins [41]. The responses for particle size, PDI, zeta potential and insulin release in simulated gastric medium after 120 min and intestinal medium after 180 min are shown in Table 1.

The nanoparticle diameter was chitosan- and albumin dependent. The minimum size (within 313–317 nm) corresponded to the highest chitosan (0.075%) and higher albumin (1.0 and 1.5%) concentrations independently of the PEG concentration, according to Table 1. Interestingly, keeping PEG and albumin concentrations fixed at 0.02 and 1.5%, respectively, the size of the nanoparticles varied as a function of the chitosan content. The same trend was obtained at prefixed PEG and albumin concentrations of 0.02 and 0.5%, respectively. In both cases, the nanoparticle size was reduced by more than 240 nm when the chitosan concentration was increased from 0.025 to 0.075%. The effect of albumin and chitosan concentrations on nanoparticle diameter can be attributed to a reduction in the electrical repulsion within nanoparticle polymer networks, since modifications in their electrical state may lead to nanoparticle swelling or shrinking [42].

Particle size distribution is a relevant characterization parameter of nanoparticles, as significant variations have been observed in the drug bioavailability and efficacy of nanoparticle formulations with broad particle size distributions [35]. Comparing formulations 9 to 11 or 10 to 12, nanoparticle size decreased in both cases, while the change in zeta potential between both formulations was │4│ and │3│ mV, respectively. It is important to note that both changes occurred in the stability region of zeta potential. The effect of PEG concentration on nanoparticle size distribution resulted in less particle aggregation, leading to a narrower size distribution. PDI was lower (0.17) when the PEG concentration was higher (0.30%), but it also depended on the chitosan and albumin concentrations. This low PDI is relevant, considering that the mean particle size and the particle size distribution can be critical factors for the evaluation of the performance of nanoparticle formulations [25,34]. Surprisingly, the PEG concentration increased the PDI at a chitosan concentration equal to 0.050% (formulations 5 and 6). Notably, in all cases, the albumin concentration was kept unchanged.

The results in Table 1 show that the zeta potential of the nanoparticles for all formulations was strongly negative. The zeta potential was lower than −29 mV for all formulations. The zeta potential values were mainly dependent on chitosan due to the protonated amino group, where the higher the concentration of chitosan was, the higher the zeta potential value. Alginate/dextran sulfate cores produced by the same protocol revealed a zeta potential of −36 mV [40], confirming the prevalence of predominantly negatively charged groups in the biopolymers alginate and dextran sulfate. The resulting nanoparticles upon coating with chitosan and PEG still did not reverse the zeta potential to a positive value, contrary to what was observed for similar nanoparticle formulations coated with more chitosan [43]. The effect of chitosan coating on the zeta potential of nanoparticles depends on the experimental conditions, including pH and the type and amount of chitosan [44,45]. Further coating of nanoparticles with albumin at pH 4.6 did not show an effect on zeta potential because at this pH, close to its isoelectric albumin is less negatively charged compared to a coating step at pH 5.1 [44]. Although albumin presented a zeta potential close to 0 mV at pH 4.6 [13], it still has protonated groups that may interact through a balance of repulsive electrostatic forces, H bonds and hydrophobic forces with nanoparticle components such as chitosan [46,47]. Lower zeta potential can be interpreted as a higher electrostatic stabilizing effect of nanoparticles in aqueous suspension, which suggests a low aggregation of nanoparticles in most of the conditions to which the nanoparticles have been exposed in this work. Zeta potential, which depends on the surface charge of nanoparticles, is essential for the stability of nanoparticles in an aqueous suspension [48] as well as a significant factor in the adsorption of nanoparticles onto the cell membrane [49]. A negative zeta potential reveals a predominance of negatively charged groups, thereby suggesting the presence of an albumin coating on the nanoparticle surface that interacts with predominantly positively charged chitosan [35]. High stability of nanoparticles in aqueous suspension is relevant, and their maintenance during manufacturing can predict better insulin physicochemical and biological stability in drug delivery systems such as nanoparticles [50].

Insulin release in simulated GI media was assessed. First, insulin retention within nanoparticles and consequent protection against acidic degradation. Later, for insulin release in simulated intestinal medium, insulin can be absorbed through the intestinal epithelium. Surprisingly, when fixing chitosan and albumin concentrations, a change in the PEG concentration did not result in the variation of insulin release in simulated gastric medium.

No insulin escaped from nanoparticles in simulated gastric medium after 120 min for formulations 3, 4, 8, and 12 with higher albumin concentrations (1.0 and 1.5%). Insulin release from nanoparticles was primarily observed for formulations with lower concentrations of albumin (0.5%) and chitosan (0.025%), where the insulin release was up to 60%, depending on the mentioned factors at the lowest level. High concentrations of albumin coatings led to lower insulin release, likely due to a strengthening of the electrostatic interaction between the positively charged albumin/chitosan network and the negatively charged alginate/dextran core reliant on the pH conditions. Insulin release from nanoparticle formulations containing a lower concentration of chitosan occurs because, at low pH, the ionic interaction between encapsulated insulin and the alginate/dextran core is weakened due to a destabilization effect by ions present in simulated gastric medium [19]. When compared to similar nanoparticle formulations, the formulation reported herein due to the chitosan/albumin coating showed high retention of insulin. Many of the previously studied nanoparticle formulations for oral delivery of insulin have not retained the peptide drug under simulated gastric conditions [13,51,52], not providing the highest amount of insulin initially encapsulated to be absorbed in the intestinal tract and thus not contributing to the highest insulin oral bioavailability. After incubation in gastric medium, nanoparticles were transferred to simulated intestinal medium. A cumulative insulin release, between 50 and 78%, was observed after 180 min for all formulations in the simulated intestinal medium. Among those, a lower value of cumulative release was observed for formulations with the highest concentration of chitosan (0.075%). Incomplete release probably occurs due to insulin–polysaccharide and insulin–albumin interactions. Therefore, the amount of insulin retained within nanoparticles under intestinal simulation may be tightly bound to the alginate nucleus, requiring more extensive dissolution for additional release. The pH triggered insulin release when incubated nanoparticles in the acidic gastric medium reached the intestinal medium. This three-hour insulin release would result in its availability close to the absorption site, which constitutes an excellent benefit for oral insulin bioavailability [19]. CD, the technique to monitor the integrity of insulin against harsh conditions [18], as seen in Figure 1, showed that the spectrum of unprocessed standard insulin (I) in PBS at pH 7.4 has bands with two minima at approximately 209 and 224 nm, indicating the presence of a significant α-helix structure with some β–sheets. Insulin released from nanoparticles (II) showed a similar spectrum. Nevertheless, minima were attenuated with respect to the standard solution. The interpretation of this result is that the secondary insulin structure may have slightly changed upon encapsulation into the nanoparticle. The simplest explanation is that the peptide drug could be linked to the biopolymers, resulting in the modification of the protein structure, although not representing denaturation or loss of insulin activity. Thus, the use of nanoparticles allows the preservation of the secondary structure of insulin after being released as a consequence of media exposure [53].

### 2.2. Fitting Data of Dependent Variables to Model Statistics

An inverse relationship depending on chitosan and albumin concentration was found in the nanoparticle formulation. The mean particle size of the nanoparticles varied from 314 to 585 nm depending on the chitosan and albumin concentrations, as shown in Table 1. The effect of levels of the independent variables on particle size is shown in Figure 2.

In Table 1, an inverse relationship between the PDI and the PEG concentration was observed in formulations prepared with chitosan and albumin concentrations of at least 0.05% and 1.0%, respectively, where the PDI decreased by increasing the PEG concentration from 0.01 to 0.03% while keeping the concentration of chitosan and albumin solutions constant, as observed in formulations 3 and 4. As nanoparticle dispersions were submitted to dialysis, the escape of PEG from the nanoparticle structure cannot be excluded. The PEG density on nanoparticles is hardly achieved since continuous and complete separation of excess polymers from nanoparticle dispersions may lead to particle aggregation and precipitation [54]. In this experimental work, the presence of PEG in the nanoparticle structure was assessed using FTIR analysis. As shown in Figure 3, nanoparticles prepared with chitosan/PEG exhibit FTIR spectra similar to those of nanoparticles prepared without PEG. Although no evidence for developing new bands or disappearance of characteristic bands considered relevant to PEG was observed, changes in the shift in the absorption bands assigned to PEG located at 962, 1278 and particularly 2885 cm^−1^ [33] were observed. This indicates the presence of an interaction between chitosan and PEG, which could proceed from the intermolecular hydrogen interactions between chitosan and PEG [55].

The zeta potential of the nanoparticle suspension varied in the range of −29 to −44 mV, as presented in Table 1. The zeta potential values were mainly dependent on chitosan due to the protonated amino group, where the higher the concentration of chitosan was, the higher the zeta potential value. The zeta potential interval may indicate the nanoparticles’ aqueous stability, with values higher than 30 mV in absolute modulus representative of stable nanoparticle formulations in suspension [48]. Insulin release in both simulated gastric medium and simulated intestinal medium depended on chitosan and albumin concentrations. Notably, insulin release from nanoparticles in simulated intestinal medium was higher for formulations with low and medium levels of chitosan, as seen in Table 1.

Experimental data were statistically analyzed, searching for the models best fitting the independent variables. Therefore, a quadratic model was established for the dependent variables’ particle size, PDI, zeta potential and insulin release in simulated gastric and intestinal media with high fitting coefficients (above 0.95). A linear model was established for the zeta potential with a fitting coefficient of 0.81. The regression equations of each model were plotted. Then, a polynomial equation comprising the individual main effect as well as the effect derived from the interaction between components was selected based on the determination of statistical parameters to optimize the nanoparticle formulation.

Table 2 shows the coefficients of all the independent variables related to their effect and their comparative significance on the responses observed in the dependent variables. In the regression equation, a positive value represents a beneficial effect on the optimization as a synergistic effect occurs, whereas a negative value represents an inverse relationship as an antagonistic effect between the independent factor and the response is likely to occur [56].

The independent variable corresponding to chitosan concentration (X_2_) negatively affected particle size (Y_1_) and PDI (Y_2_) responses and released insulin from nanoparticles in simulated GI media (Y_4_ and Y_5_), whereas a positive effect on zeta potential (Y_3_) was observed. Albumin (X_3_) negatively minimized insulin release from nanoparticles in simulated gastric medium (Y_4_). In contrast, the PEG (X_1_) concentration negatively affected the PDI (Y_3_). The PDI was lower at a higher PEG level, possibly due to a lower tendency of multilayer complexes to form aggregates.

Regression equation of the fitted model:(2)$$Y=b_{0}+b_{1}X_{1}+b_{2}X_{2}+b_{3}X_{3}+b_{12}X_{1}X_{2}+b_{13}X_{1}X_{3}+b_{23}X_{2}X_{3}+b_{11}X_{1}^{2}+b_{22}X_{2}^{2}+b_{33}X_{3}^{2}$$

Higher-order terms or coefficients with more than one factor in the obtained regression equation correlate to a quadratic relationship or an interaction between terms, respectively, suggesting a nonlinear relationship between independent and dependent variables [31]. In this way, independent variables can originate different degrees of response when compared to that predicted by regression equations upon their variation at different levels or in case of simultaneous changes of more than one factor. Except for zeta potential, for which independent variables presented a linear relationship, responses in Y_1_, Y_2_, Y_4_ and Y_5_ were affected by the interactions between factors, demonstrating a quadratic relationship. The interaction effect between X_1_ and X_3_ showed a negative effect on PDI and was twofold higher than the effect of X_1_. The interaction effect between X_2_ and X_3_ was favorable for response in Y_4_ but did not affect response in Y_5_. The quadratic effects of X_2_ and X_3_ were observed for responses in Y_1,_ Y_2_, Y_4_ and Y_5_, whereas most positive quadratic effects for X_2_ and X_3_ were observed for Y_1_ and a negative quadratic effect for X_2_ was observed for Y_5_.

### 2.3. Response Surface Analysis

Graphs of three-dimensional models are plotted in Figure 4, Figure 5, Figure 6 and Figure 7, in which response analyses have been plotted toward optimization of the critical dependent variables of nanoparticles for oral insulin delivery. Response surface plots can be used to interpret the interaction effects of two independent variables on the dependent variables when a third factor is kept at a constant level. Except for the zeta potential, where the interaction effects of PEG and chitosan were linear, the relationships among the three independent variables were nonlinear.

Small particle sizes are most likely to increase intimate contact with the intestinal mucosa, as their higher surface area-to-volume ratio increases nanoparticle uptake in GI mucosa. As seen in Figure 4, a more pronounced effect of chitosan concentration on nanoparticle size is observed for chitosan concentration values lower than 0.05%, whereas, lower PDI values were obtained with a higher concentration of PEG and an intermediate and higher concentration of chitosan, as seen in Figure 5.

Insulin release in enzyme-free simulated digestive media depended on chitosan and albumin concentrations, as shown in Figure 6 and Figure 7. The protection of insulin against adverse conditions in gastric simulation through its retention within the nanoparticles is obtained by a higher concentration of chitosan and albumin at a constant level of PEG.

Upon transferring nanoparticle formulations into the intestinal medium, the insulin release from nanoparticles increases when the chitosan concentration is lower than 0.05%, regardless of the albumin concentration tested, as shown in Figure 6.

### 2.4. Optimization and Model Validation

The optimum nanoparticle formulation can be set by analyzing various dependent variables and monitoring the constraints by a mathematical approach. Following the constraints of the parameters established in the Box–Behnken design, the optimum nanoparticle formulation comprising biopolymers for insulin delivery by the oral route was selected. It is formulated with solutions of 0.03% PEG, 0.047% chitosan and 1.20% bovine serum albumin and has predictive values of particle size of 357 nm, PDI of 0.19, zeta potential of −35.0 mV, total retention of insulin in gastric conditions and 76% release in the simulated intestinal medium, as presented in Table 3.

The formulation of nanoparticles according to the composition stated in Section 2.4 validates the Box–Behnken design obtained in this work since dependent variables showed experimental values with an error equal to or lower than 5% with respect to predicted ones, as seen in Table 3. The optimized nanoparticle formulation has a mean particle size of 357 nm, PDI of 0.19, zeta potential of −35 mV, full insulin retention within nanoparticles in the enzyme-free simulated gastric medium for 120 min, and insulin release equals 76% in enzyme-free intestinal simulation after 180 min.

## 3. Conclusions

The Box–Behnken design was applied to optimize the formulation of nanoparticles and to evaluate the main interaction of the factors that influence the formulation obtained. The quadratic effects of these factors on particle size, PDI, zeta potential, and insulin release from nanoparticles in simulated gastric and intestinal media were also studied. Experimental designs allowed the multiparametric optimization of the nanoparticle formulation by selecting physicochemical parameters critical for oral delivery of insulin, evaluating the most relevant factors on responses, and investigating any relationship existing between factors upon response surface methodology. A 3-factor, 3-level design based on 15 experiments allowed for exploring the linear and quadratic response surfaces and establishing a second-order polynomial model. Chitosan and albumin, as coating biopolymers, were revealed to be the main formulation factors regarding the desired physicochemical properties of the nanoparticles, except for PDI and insulin release under simulated GI conditions. Based on the experimentally obtained values and according to desirability, solutions of 0.03% PEG, 0.047% chitosan and 1.2% albumin led to the optimum nanoparticle formulation for oral insulin delivery. Compared to previous nanoparticle-based formulations prepared using the same protocol, the factorial optimized formulation resulting from this work showed a narrow size distribution induced by the PEG/chitosan ratio. In addition, the model developed in this work increases nanoparticle characterization robustness, thereby making it easier to predict nanoparticle properties such as drug release, blood circulation time, bioavailability and cellular uptake.

## 4. Materials and Methods

### 4.1. Materials

Alginic acid sodium salt (200 kDa with a mannuronic/guluronic ratio of 1.56, Ref A2158), low molecular weight chitosan (50 kDa with a deacetylation degree >75%, Ref. 448869), bovine serum albumin (66.5 kDa Ref A1933) and trifluoroacetic acid (TFA) 99% (*v*/*v*) were purchased from Sigma–Aldrich (Madrid, Spain), dextran sulfate sodium salt (5 kDa) and polyvinylpyrrolidone (PVP) K 30 were purchased from Fluka (Buchs, Switzerland), poloxamer 188 (Lutrol^®^ F68) was kindly supplied by BASF (Hürth, Germany), calcium chloride and sodium chloride were purchased from Riedel-de-Haën (Lower Saxony, Germany), lactic acid 90% was purchased from VWR BDH Prolabo (Rosny-sous-Bois, France), polyethylene glycol 4000 (PEG 4000) was acquired from Fisher Scientific^®^ (Loughborough, UK), acetonitrile LiChrosolv^®^, hydrochloric acid 37%, potassium dihydrogen phosphate and sodium hydroxide were purchased from Merck KGaA (Darmstadt, Germany), and Actrapid^®^ 100 IU/mL (Novo Nordisk A/S, Bagsværd, Denmark) was supplied by a local pharmacy. Biopolymer solutions were prepared in ultrapure water. Chitosan was dissolved in an aqueous solution containing lactic acid at 0.5% (*v*/*v*), and otherwise stated solutions were under-vacuum filtered using a Millipore#2 paper filter.

### 4.2. Methods

#### 4.2.1. Preparation of Nanoparticles

Nanoparticles were prepared using ionotropic pregelation [57] of alginate/dextran sulfate solution containing poloxamer 188 and insulin with calcium ions, following polyelectrolyte complexation with both oppositely charged chitosan and albumin.

Ionotropic pregelation involved dropwise extrusion of 7.5 mL of a calcium chloride solution into 117.5 mL of pH 4.9 0.06% (*w*/*v*) alginic sodium salt, 0.04% (*w*/*v*) dextran sulfate, 0.04% (*w*/*v*) poloxamer 188 and 0.006% (*w*/*v*) insulin at constant stirring. A two-step complexation involved dropwise addition of 25 mL of chitosan and polyethylene glycol 4000 solution at pH 4.6 for stabilization of the pregel core into nanoparticles, followed by dropwise addition of 25 mL bovine serum albumin solution at pH 4.6. The concentration for each of the last three components varied between formulations, as indicated in Table 4. Nanoparticles were concentrated after pregelation and coating steps by dialysis [39] using a regenerated cellulose membrane with a tubing nominal dry thickness of 10 kDa molecular weight cutoff (MWCO) (SnakeSkin Pleated Dialysis Tubing, Thermo Fisher Inc., Waltham, MA, USA) and a dialysis solution of 20% (*w*/*v*) PVP K 30 for 24 h at 4 °C. The pH of the suspension was set at 4.9, and KNO_3_ as an ionic agent was added at 0.075% (*w*/*v*) [13].

#### 4.2.2. Particle Size Analysis

Nanoparticle size was characterized by using dynamic light scattering (DLS) (NanoZetasizer, Malvern, UK) at 25 °C with a detector angle of 173°, setting the nanoparticle concentration based on a suitable operating procedure (SOP) of the instrument. Measurements were made in hexaplicate.

The number of runs was established by the software to reach the quality criteria. Each run lasted 10 s with no delay between measurements. Nanoparticle formulations were screened for one-week size stability of samples upon refrigeration of samples between 2–8 °C [13]. For this study, size measurements were carried out after preparation and refrigeration of samples between 2–8 °C. Each curve in a plot shows the average of the measurements using a protocol validated for reproducible intensity and number distributions. Distribution by intensity allowed the characterization of nanoparticle size. In contrast, distribution by number was obtained by the software assuming the particles to be spherical, the homogeneity of the sample, and the accuracy of the distribution by intensity, allowing the relative populations of the particles to be estimated.

#### 4.2.3. Zeta Potential Analysis

The potential ζ, an electrical charge-related measurement on the surface of a nanoparticle, was performed by using the same apparatus. For each assay, three automated measurements were made.

#### 4.2.4. Insulin Release Studies

Insulin release from the nanoparticles was determined in simulated enzyme-free digestive media. A sample of 3 mL was added into dialysis diffusion bags with an MWCO of 100 kDa (Spectra/Por^®^, Biotech CE, Spectrum Laboratories Inc., Piscataway, CA, USA) and then immersed in 100 mL of simulated pepsin-free gastric medium [58] at 37 °C (120 min/100 rpm), followed by incubation in a simulated pancreatin-free intestinal medium [58] for 180 min after recovering nanoparticles by centrifugation (20,000× *g*/15 min). Sample aliquots were withdrawn after 120 min in gastric medium and 180 min after transferring nanoparticle formulations to intestinal medium conditions.

Release studies were carried out in enzyme-free media to determine the pH-responsive properties of nanoparticles, minimizing interference of enzymes that may not reveal changes toward the pH shift from the stomach to the small intestine. The nanoparticles analyzed varied in the concentration-tested ranges of PEG, chitosan and albumin. Collected samples were submitted to centrifugation (20,000× *g*/15 min), and the supernatant was assayed for insulin by HPLC. The cumulative percentage release of insulin from nanoparticles refers to the insulin content in the nanoparticles. Studies were carried out in triplicate.

#### 4.2.5. Insulin Determination

The determination of insulin was performed using an LC-2010 HT HPLC system (Shimadzu Co., Kyoto, Japan) equipped with a quaternary pump, an HP 1050 programmable multiple wavelength detector set at 214 nm, a reversed-phase X-Terra^®^ RP 18 column, 5 lm, 4.6 mm × 250 mm (Waters Co., Milford, MA, USA) and a Purospher STAR^®^ RP-18 precolumn 5 µm (Merck KGa, Darmstadt, Germany). A gradient-operated mobile phase consisting of acetonitrile (A) and 0.1% trifluoroacetic acid (TFA) aqueous solution (B) at a flow rate of 1.0 mL/min set to 30:70 (A:B), changed to 40:60 (A:B) in 5 min for elution over 5 min, and changed to 30:70 (A:B) in 1 min for elution over 1 min. Peak area responses of the chromatograms were measured with an automatic integrator. The method was validated and was linear in the range of 2.1–108 µg/mL (R^2^ = 0.9996).

#### 4.2.6. Conformational Stability of Insulin

By using circular dichroism (CD) spectroscopy, the secondary structure of insulin released from the nanoparticles was evaluated. The CD spectra were collected using a Jasco J-815 spectropolarimeter (Tokyo, Japan) with a temperature controller. Spectra were collected at 25 °C using a 0.1 cm cell over a 200–260 nm wavelength range. A resolution of 0.2 nm and scanning speed (50 nm/min) with a 4-s response time was employed. Each spectrum acquired is an average of five consecutive scans. Blank buffer subtraction, noise reduction and data analysis were performed using Jasco’s standard and temperature/wavelength analysis software. The spectra of insulin samples extracted from nanoparticles with concentrations of approximately 10 µM in phosphate-buffered saline (PBS) were compared with those of unprocessed insulin in the same medium.

#### 4.2.7. Fourier Transform Infrared (FTIR) Spectroscopy

FTIR analysis was used to ascertain the presence of PEG in the nanoparticle structure. Infrared spectra of freeze-dried nanoparticle formulations and PEG were recorded in the range of 650 to 4000 cm^−1^ with a 400 N FT-NIR Imaging System (Perkin-Elmer, Tampa, FL, USA). Each sample was read in 64 scans at a resolution of 4 cm^−1^. The formulations were frozen overnight at −80 °C and dried in a chamber at 0 °C for 48 h at 0.133 mbar, corresponding to a condenser temperature of −50 °C, using a Lyph-lock 6 apparatus (HETO LyoPro 3000, Heto/Holten A&S, Allerød, Denmark).

#### 4.2.8. Experimental Design

The Box–Behnken design was selected because it requires a low number of runs in the case of three variables. As seen in Table 4, a 3-factor, 3-level design was used to optimize nanoparticle formulation with PEG and chitosan and albumin concentrations, which were defined as the independent variables or formulation factors with three levels of concentration values, low, medium and high. The points located at the median values of the edges of the experimental design were evaluated in triplicate [59]. These center points are useful to determine if there is curvature in the relationship between independent and dependent factors. In addition, using several center points enables an estimate of pure error. The range of concentrations was established based on previous studies of similar nanoparticles containing insulin [13,39], where PEG was determined to be appropriate for promoting particle stability, and the chitosan amount strengthened the alginate/dextran sulfate while interacting with another polyelectrolyte polymer, albumin, which proved critical as a sacrificial target, thus protecting insulin within the nanoparticle. The dependent variables are nanoparticle size, PDI, zeta potential, and insulin release in simulated GI media after 120 min and 180 min, with disclosure of constraints applied, as described in Table 4. Design-Expert^®^ software (v.13 Stat Ease Inc., Minneapolis, MN, USA) was used to generate and evaluate the statistical experimental design. The values of formulation factors and the corresponding responses for these dependent variables are shown in Table 4.

#### 4.2.9. Analysis of Experimental Data and Model Validation

Polynomial equations for disclosure of the main effect and interaction among factors were determined upon estimating statistical parameters, including multiple correlation coefficients, adjusted multiple correlation coefficients and the predicted residual sum of squares generated by the software. To determine the optimized formulation, three-dimensional surface plots were drawn. All responses were fitted to linear or quadratic models. The polynomial equations’ validation was established by analysis of variance (ANOVA) provision available in the software. Accordingly, the optimum values of the dependent variables were determined graphically and numerically using Design-Expert^®^ and based on the criterion of desirability [60].

Following the preparation of nanoparticles according to the optimum formulation, the resultant experimental responses were compared with the predicted responses to determine the percentage of the predicted error [59]. The optimization protocol was validated for predicted error values lower than 5%.

#### 4.2.10. Statistical Analysis

Measured values are represented as the mean ± standard deviation (SD) of at least three independent experiments. One-way ANOVA with Bonferroni post hoc test (SPSS 20.0, Chicago, IL, USA) was used to statistically analyze the data. The level of significance was set at probabilities of * *p* < 0.05, ** *p* < 0.01, and *** *p* < 0.001.

## Figures and Tables

**Figure 1 marinedrugs-21-00179-f001:**
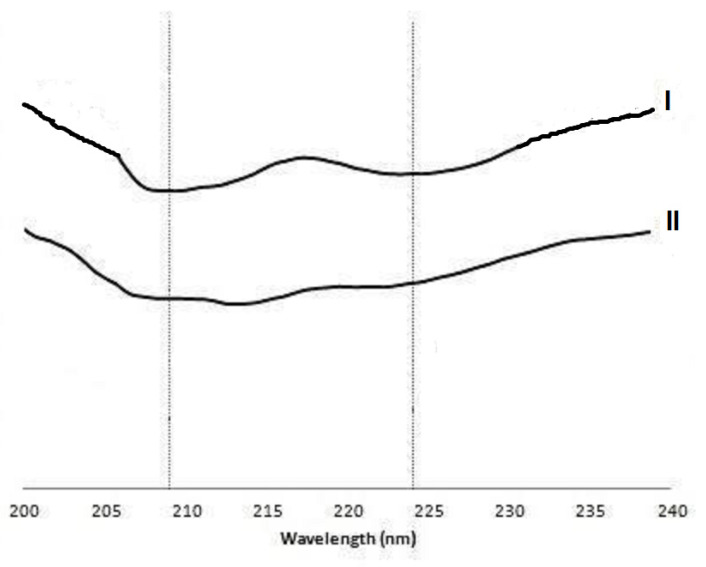
Circular dichroism (CD) spectra of insulin in solution (10 µM) in phosphate-buffered saline (PBS) at pH 7.4 and 25 °C: (I) standard solution (unprocessed), (II) insulin released from nanoparticles.

**Figure 2 marinedrugs-21-00179-f002:**
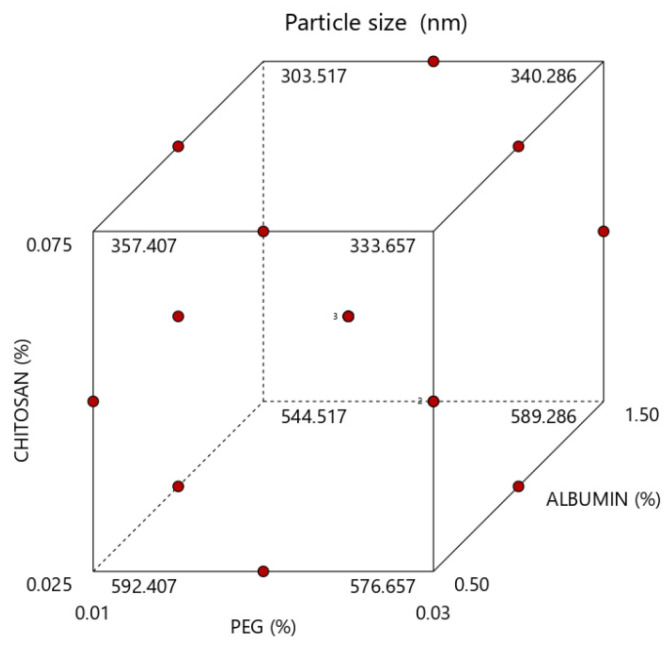
Model graph representing the effect of formulation factors on particle size.

**Figure 3 marinedrugs-21-00179-f003:**
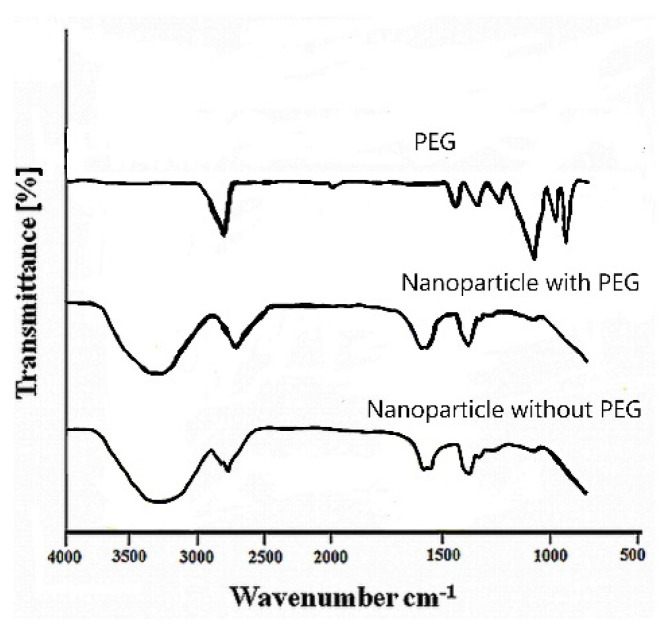
FTIR spectra of pure PEG and chitosan/albumin-coated alginate/dextran sulfate nanoparticles with and without PEG.

**Figure 4 marinedrugs-21-00179-f004:**
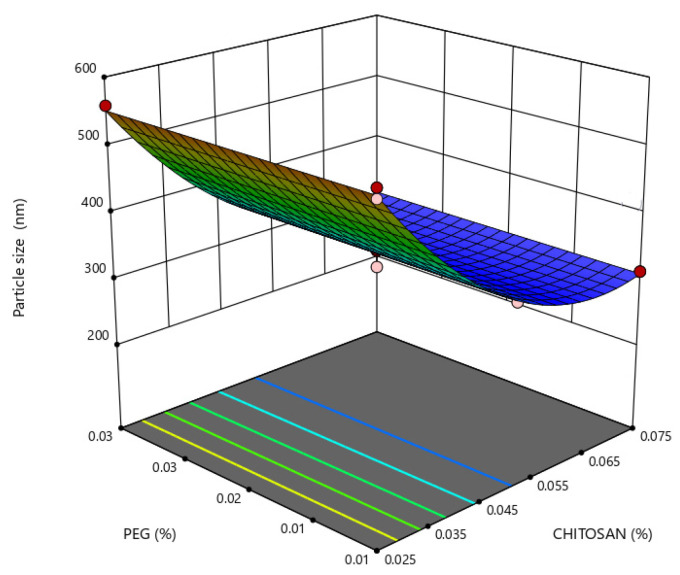
Response surface plot representing the effect of polyethylene glycol (PEG) (X_1_) and chitosan (X_2_) concentrations on nanoparticle size (Y_1_).

**Figure 5 marinedrugs-21-00179-f005:**
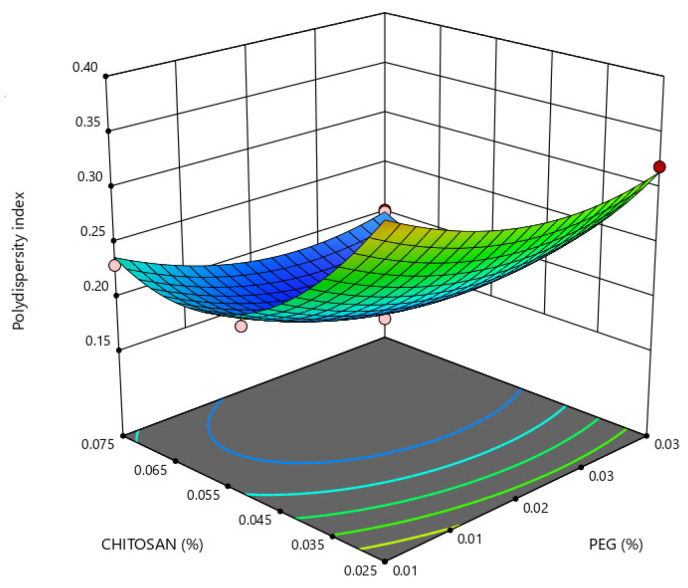
Response surface plot representing the effect of polyethylene glycol (PEG) (X_1_) and chitosan (X_2_) concentrations on nanoparticle polydispersity (Y_2_) when albumin concentration is at a constant level of 0.50% (X_3_).

**Figure 6 marinedrugs-21-00179-f006:**
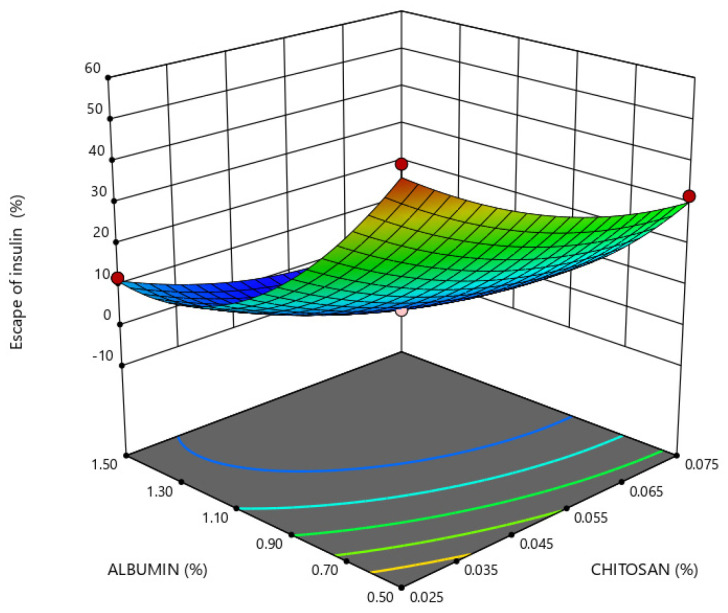
Response surface plot representing the effect of chitosan (X_2_) and albumin (X_3_) concentrations on insulin escape from nanoparticles in simulated gastric medium (Y_4_) when the polyethylene glycol (PEG) concentration is at a constant level of 0.02% (X_1_).

**Figure 7 marinedrugs-21-00179-f007:**
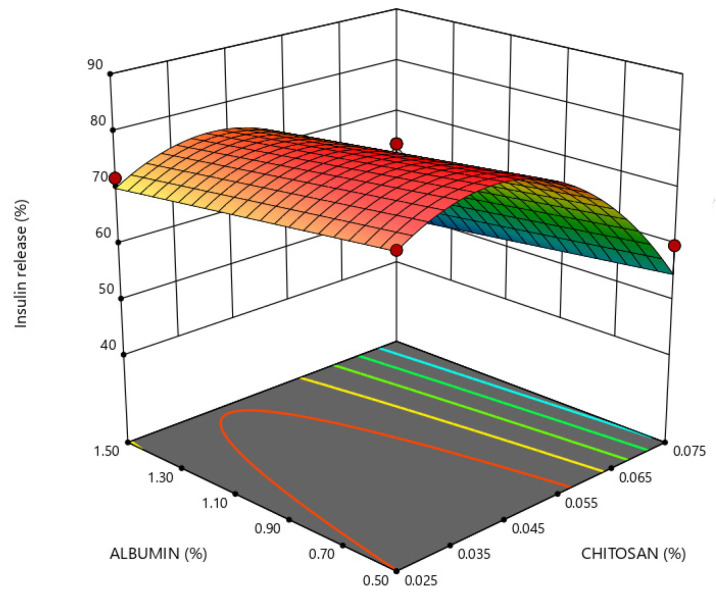
Response surface plot representing the effect of chitosan (X_2_) and albumin (X_3_) concentrations on cumulative insulin release in simulated intestinal medium (Y_5_) when the polyethylene glycol (PEG) concentration is at a constant level of 0.02% (X_1_).

**Table 1 marinedrugs-21-00179-t001:** Formulation factors and observed responses in the Box–Behnken experimental design.

Formulation	Independent Variables			Dependent Variables				
	X_1_ (%)	X_2_ (%)	X_3_ (%)	Y_1_ (nm) mean ± SD	Y_2_ mean ± SD	Y_3_ mV ± SD	Y_4_ % ± SD	Y_5_ %± SD
1	0.01	0.025	1.0	547 ± 21	0.36 ± 0.04	−43.0 ± 1.3	16 ± 2	74 ± 2
2	0.03	0.025	1.0	559 ± 23	0.32 ± 0.02	−41.0 ± 1.5	19 ± 3	70 ± 3
3	0.01	0.075	1.0	313 ± 13	0.23 ± 0.02	−31.0 ± 0.7	0	50 ± 4
4	0.03	0.075	1.0	317 ± 15	0.20 ± 0.03	−30.0 ± 0.8	0	52 ± 6
5	0.01	0.050	0.5	378 ± 20	0.22 ± 0.04	−34.0 ± 1.5	37 ± 3	78 ± 3
6	0.03	0.050	0.5	362 ± 17	0.27 ± 0.04	−31.0 ± 1.3	35 ± 4	76 ± 3
7	0.01	0.050	1.0	339 ± 14	0.22 ± 0.03	−36.0 ± 0.7	6 ± 4	76 ± 5
8	0.03	0.050	1.5	376 ± 13	0.17 ± 0.01	−34.0 ± 0.8	0	76 ± 3
9	0.02	0.025	0.5	585 ± 13	0.27 ± 0.03	−44.0 ± 1.8	60 ± 5	76 ± 4
10	0.02	0.075	0.5	342 ± 15	0.27 ± 0.02	−32.0 ± 0.8	32 ± 4	60 ± 3
11	0.02	0.025	1.5	563 ± 24	0.39 ± 0.05	−40.0 ± 1.2	12 ± 3	72 ± 4
12	0.02	0.075	1.5	314 ± 14	0.18 ± 0.02	−29.0 ± 0.7	0	46 ± 6
13 *	0.02	0.050	1.0	320 ± 26	0.19 ± 0.02	−34.0 ± 2.2	6 ± 2	78 ± 3
14 *	0.02	0.050	1.0	345 ± 14	0.18 ± 0.03	−33.0 ± 2.1	5 ± 3	76 ± 4
15 *	0.02	0.050	1.0	347 ± 13	0.19 ± 0.01	−33.0 ± 1.1	4 ± 2	78 ± 4

* Defined as a center point.

**Table 2 marinedrugs-21-00179-t002:** Coefficients of the regression equation for formulation factors as independent variables and standard error.

Terms	Y_1_			Y_2_			Y_3_			Y_4_			Y_5_		
	C	SE	Range *	C	SE	Range *	C	SE	Range *	C	SE	Range *	C	SE	Range *
b_0_	341.19	5.01	330.04 to 332.35	0.185	0.006	0.170 to 0.200	−35	0.53	−36 to −34	4.14	0.93	2.02 to 6.25	76.46	1.01	74.23 to 78.69
b_1_	-	-	-	−0.02	0.004	−0.03 to −0.01	-	-	-	-	-	-	-	-	
b_2_	−121.00	4.03	−129.97 to −112.03	−0.06	0.004	−0.07 to −0.05	5.75	0.73	4.17 to 7.33	−9.38	0.76	−11.07 to −7.68	−19.50	0.94	−12.56 to −8.44
b_3_	-	-	-	-	-		-	-	-	−18.98	0.82	−20.82 to −17.13	-	-	
b_1_b_2_	-	-	-	-	-		-	-	-	-	-	-	-	-	
b_1_b_3_	-	-	-	−0.05	0.007	−0.07 to−0.03	-	-	-	-	-	-	-	-	
b_2_b_3_	-	-	-	−0.05	0.005	−007 to −0.04	-	-	-	4.00	1.06	1.60 to 6.40	-	-	
b_1_^2^	-	-	-	0.03	0.006	0.01 to 0.04	-	-	-	-	-		-	-	
b_2_^2^	89.36	5.95	76.11 to 102.61	0.07	0.006	0.05 to −0.08	-	-	-	5.73	1.11	3.22 to 8.24	−13.96	1.38	−17.00 to −10.92
b_3_^2^	23.88	5.95	−10.64 to 37.14	0.03	0.006	0.01 to 0.05	-	-	-	15.02	1.11	12.51 to 17.53	-	-	

C: coefficient estimate; SE: standard error. * The range indicates the lower and upper values of coefficients at 95% confidence interval.

**Table 3 marinedrugs-21-00179-t003:** Comparison between predicted and experimental responses and predicted error for optimized nanoparticle formulation.

Dependent Variable	Predicted Response	Experimental Response	Predicted (%)
Particle size (nm)	357	366	+2.5
Polydispersity index (PDI)	0.193	0.202	+4.4
Zeta potential (mV)	−35	−34	+2.9
Insulin escape in gastric medium (%)	0	0	0
Intestinal release in intestinal medium (%)	76	73	−5.0

**Table 4 marinedrugs-21-00179-t004:** Dependent and independent variables in Box–Behnken 3-level 3-factor design of nanoparticle formulations.

Parameter	Levels		
Independent variables	−1	0	1
Polyethylene glycol (PEG) % (*w*/*v*)	0.01	0.02	0.03
Chitosan % (*w*/*v*)	0.025	0.050	0.075
BSA %(*w*/*v*)	0.5	1.0	1.5
Dependent variables	Constraint		
Particle size (nm)	Minimize		
Polydispersity index (PDI)	Minimize		
Zeta potential (mV)	Less than −30 mv		
Insulin escape in gastric medium (%)	Minimize		
Intestinal release in intestinal medium (%)	Maximize		

## Data Availability

The data presented in this study are available in the present article.

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
