# Peer review of "Chitosan/Albumin Coating Factorial Optimization of Alginate/Dextran Sulfate Cores for Oral Delivery of Insulin"

_marinedrugs, 2023, doi:10.3390/md21030179_

Round 1

Reviewer 1 Report

Pessoa et al. reported the methodology of using a 3-factor 3-level optimization Box-Behnken design to predict the optimized parameters to synthesize a nanoparticle. The settings of dependent variables were reasonable, and the authors precisely predicted the optimized parameters based on the methodology. Although the methodology works efficiently and might be applied to other applications, the novelty is somehow questionable. I also remain concern about the synthesis protocol/step.

l   The composition of the nanoparticle reported in the manuscript has been reported years ago (lonso-Sande et al., Macromolecules, vol. 39, no. 12, pp. 4152–4158, 2006 and Reis et al. Carbohydr. Polym., vol. 76, no. 3, pp. 464–471, 2009), and the Box-Behnken design for similar nanoparticles were reported too (Arora et al., Sci Pharm. 2011 Jul-Sep; 79(3): 673–694). The authors have to address the novelty and importance of the manuscript before to be considered published in the journal.

l   Page 3, methods for nanoparticle preparation. Was the nanoparticles complexation purified (e.g., dialysis or centrifugation) before the addition of the solutions containing coating materials? Or was it a one-pot synthesis?

l   Why the BSA coating was performed under pH 4.6, where happened to be the isoelectric point of BSA?

l   In page 9, line 348, the authors explained that the value of zeta potential was highly dependent to the chitosan concentration due to the protonated amino groups. Did the authors try to express that the higher concentration of chitosan cause higher BSA coating on the surface of the nanoparticle? Also, it seemed that the size

Minor

l   Page 2 line 58, the “tendency to aggregate” in the sentence might cause confusion to the readers. Please revise the sentence.

l   Page 2, line 74, why did the author mentioned zein, as zein was not used as a component in the nanocomplex? The author can consider removing it or add context to address. It.

l   Page 6, line 236, please define what is a “center point”?

Author Response

Responses to Reviewer #1

Pessoa et al. reported the methodology of using a 3-factor 3-level optimization Box-Behnken design to predict the optimized parameters to synthesize a nanoparticle. The settings of dependent variables were reasonable, and the authors precisely predicted the optimized parameters based on the methodology. Although the methodology works efficiently and might be applied to other applications, the novelty is somehow questionable. I also remain concern about the synthesis protocol/step.

l   The composition of the nanoparticle reported in the manuscript has been reported years ago (lonso-Sande et al., Macromolecules, vol. 39, no. 12, pp. 4152–4158, 2006 and Reis et al. Carbohydr. Polym., vol. 76, no. 3, pp. 464–471, 2009), and the Box-Behnken design for similar nanoparticles were reported too (Arora et al., Sci Pharm. 2011 Jul-Sep; 79(3): 673–694). The authors have to address the novelty and importance of the manuscript before to be considered published in the journal.

R: There is no novelty in the preparation protocol of nanoparticles even compared to some of our published works [1] as discriminated on page 2, line 68-72 of the revised manuscript. The composition of these nanoparticles is different from [2], and regarding the cited work by Reis, the authors could not find it. The eventual similarity with [3] is not confirmed; there are several differences in the Box Behnken design, among which:

  • The drug is not the same, and evident differences in molecular weight and stability exist;
  • Independent and dependent variables.

The novelty of this manuscript is focused on the multiparametric approach in the coating of insulin-loaded alginate/dextran sulfate cores to protect insulin stability and provide appropriate release of this peptidic drug in the GI tract as described on page 3, lines 123-127. There are other published works dealing with insulin-loaded nanoparticles prepared upon the QbD approach for oral delivery [4,5] and wound healing [6], but none have prepared nanoparticles by ionotropic gelation/complexation focused on the chitosan and albumin dual coating step.

To our knowledge, the use of Box‒Behnken design to optimize the preparation process of chitosan/albumin dual coating of alginate/dextran sulfate cores for oral delivery of insulin has limited applications. Consequently, the present work aimed to improve the physicochemical stability and insulin release in the GI tract of chitosan/albumin dual-coated alginate/dextran sulfate cores using ionotropic gelation and complexation methods and optimized by Box‒Behnken design. The authors believe that the novelty of this manuscript was not clear enough and therefore decided to make some changes, as can be found in red text on page 3, lines 123-124 of the revised manuscript.

l   Page 3, methods for nanoparticle preparation. Was the nanoparticles complexation purified (e.g., dialysis or centrifugation) before the addition of the solutions containing coating materials? Or was it a one-pot synthesis?

R: Nanoparticles were submitted between steps to dialysis against a 10 kDa membrane-containing aqueous solution. Coating alginate/dextran sulfate cores was a two-step procedure, first chitosan coating followed by an albumin coating. Changes were made in the protocol to give more details and improve the reader's understanding, as can be found in red text on page 4, lines 166-167 of the revised manuscript.

l   Why the BSA coating was performed under pH 4.6, where happened to be the isoelectric point of BSA?

R: BSA coating was performed at pH 4.6, as it has been revealed to be the optimal value for insulin protection in nanoparticles [1]. Electrostatic interactions drive the interaction of BSA with nanoparticle components. Due to the electrostatic interactions' role, the polymers' potential in the suspension is of utmost importance and, consequently, was obtained experimentally. It is essential to note that BSA in solution at pH 5.1, as in the standard formulation, had a potential close to 0 mV. Therefore, no electrostatic repulsive forces would appear on the surface of the nanoparticles, and the aggregation of the entire structures is promoted, as previously proposed[7]. Nevertheless, the reduction of the pH of the BSA suspension to 4.6 increased the zeta potential, and thus, electrostatic repulsive forces appeared on the surface of the BSA-coated nanoparticles, avoiding the interaction among them. BSA changed its conformation when reducing the pH of the solution, as the interaction with the nanoparticles and their resulting stability depend on the pH of the BSA solution.

l   In page 9, line 348, the authors explained that the value of zeta potential was highly dependent to the chitosan concentration due to the protonated amino groups. Did the authors try to express that the higher concentration of chitosan cause higher BSA coating on the surface of the nanoparticle? Also, it seemed that the size

R: Authors were expecting a BSA role on zeta potential according to previous experiments. However, these experiments did not confirm this hypothesis. The reduction of the pH of BSA to 4.6 may have increased the zeta potential [1]; therefore, electrostatic repulsive forces appeared on the surface of the BSA-coated nanoparticles. As BSA is not negatively charged under the experimental pH conditions, higher attraction to chitosan-protonated amino groups is not confirmed.

l   Page 2 line 58, the "tendency to aggregate" in the sentence might cause confusion to the readers. Please revise the sentence.

R: The authors appreciate the reviewer’s comment. The sentence was corrected by removing "tendency to aggregate" as it can be checked on page 2, line 77 of the revised manuscript.

l   Page 2, line 74, why did the author mentioned zein, as zein was not used as a component in the nanocomplex? The author can consider removing it or add context to address. It.

R: The authors acknowledge the Reviewer’s comments, and effectively, the cited reference was misused. Therefore, this reference was replaced by another one [8].

l   Page 6, line 236, please define what is a "center point"?

R: To maintain structure and coherence, authors defined the center point in methods as it can be found in red text on page 6, lines 242-244 of the revised manuscript.

References

[1]       Collado-González M, Ferreri MC, Freitas AR, et al. Complex Polysaccharide-Based Nanocomposites for Oral Insulin Delivery. Mar Drugs. 2020;18:55.

[2]       Alonso-Sande M, Cuña M, Remuñán-López C, et al. Formation of new Glucomannan - Chitosan nanoparticles and study of their ability to associate and deliver proteins. Macromolecules. 2006;39:4152–4158.

[3]       ARORA S. Amoxicillin Loaded Chitosan–Alginate Polyelectrolyte Complex Nanoparticles as Mucopenetrating Delivery System for H. Pylori. Sci Pharm. 2011;79:673–694.

[4]       Agrawal AK, Urimi D, Harde H, et al. Folate appended chitosan nanoparticles augment the stability, bioavailability and efficacy of insulin in diabetic rats following oral administration. RSC Adv. 2015;5:105179–105193.

[5]       Agrawal AK, Kumar K, Swarnakar NK, et al. "liquid Crystalline Nanoparticles": Rationally Designed Vehicle to Improve Stability and Therapeutic Efficacy of Insulin Following Oral Administration. Mol Pharm. 2017;14:1874–1882.

[6]       Dawoud MHS, Yassin GE, Ghorab DM, et al. Response Surface Optimization and In-vitro Evaluation of Sustained Release Topical Insulin Liposomal Spray for Wound Healing. J Appl Pharm Sci. 2018;8:22–29.

[7]       Valle-Delgado JJ, Molina-Bolívar JA, Galisteo-González F, et al. Interactions between bovine serum albumin layers adsorbed on different substrates measured with an atomic force microscope. Phys Chem Chem Phys. 2004;6:1482–1486.

[8]       Luo Y, Zhang B, Whent M, et al. Preparation and characterization of zein/chitosan complex for encapsulation of α-tocopherol, and its in vitro controlled release study. Colloids Surf B Biointerfaces. 2011;85:145–152.

Reviewer 2 Report

The paper  (Chitosan/Albumin Coating Factorial Optimization of Algi- 2 nate/Dextran Sulfate Cores for Oral Delivery of Insulin) .

The following points required explanations :

1-In   Materials and Methods section ( Materias ) Molecular formula and molecular weight for all chemicals used  should be written

2-In  Materials and Methods section ( Preparation of nanoparticles ) the type of complexation that involved between  chitosan and polyethylene glycol must be written

3-In  Particle Size Analysis section  The following analysis ( X-ray diffraction and Transimision electron microscopy should be done to confirm the size of particles 

4-In references part  I suggest that more references should be updated

5- English language and style are minor and  check required.

Author Response

Responses to Reviewer #2

The paper  (Chitosan/Albumin Coating Factorial Optimization of Algi- 2 nate/Dextran Sulfate Cores for Oral Delivery of Insulin) .

The following points required explanations:

1-In   Materials and Methods section ( Materias ) Molecular formula and molecular weight for all chemicals used  should be written

R: The authors acknowledge the reviewer’s observation, as it can increase the rigor of the manuscript. The molecular formula and molecular weight of biopolymers and drug, whenever possible, were discriminated, as can be found on page 3, lines 139-143, of the revised manuscript.

2-In  Materials and Methods section ( Preparation of nanoparticles ) the type of complexation that involved between  chitosan and polyethylene glycol must be written

R: The authors acknowledge the reviewer’s comments. The authors made great efforts toward disclosure of experimental conditions. Alginate/dextran sulfate cores were coated with a solution containing chitosan and PEG, as detailed in the protocol on page 4, lines 162-163 of the revised manuscript.

3-In  Particle Size Analysis section  The following analysis ( X-ray diffraction and Transimision electron microscopy should be done to confirm the size of particles

R: Our group has great expertise in nanoparticle size measurements, and our experience with dynamic light scattering is very positive. Even though a second technique would be an extra, we do not have cryo transmission electron microscopy (cryo-TEM) expertise on X-ray diffraction. We have performed scanning electron microscopy, but some artifacts may come up due to the required preparation of samples involving a drying step, which often does not simulate naturally hydrated biopolymer nanoparticles. Shortly, we have SEM images which, according to the authors, do not increase the manuscript's value.

4-In references part  I suggest that more references should be updated

R: Authors added literature references whenever they found appropriate. Criteria were the similarity of published works, including encapsulated drug, polymers, methodology, preparation based on QbD, and merit of published work. Upon Reviewer comments, the authors reviewed the introduction section, and extra references were added to the revised manuscript as follows:

Page 1, lines 30, 33, 37, 38 and 40.

Page 2, lines 51 and 95,

Page 3, lines 108 and 12

5- English language and style are minor and  check required.

R: As the Reviewer did not point out specific aspects to check, the authors read carefully throughout the manuscript, and changes were made whenever appropriate.

Reviewer 3 Report

The same nanoparticle design has been published in J Control Release, 2016 Jun 28;232:29-41. doi: 10.1016/j.jconrel.2016.04.012. Epub 2016 Apr 10. Dual chitosan/albumin-coated alginate/dextran sulfate nanoparticles for enhanced oral delivery of insulin.

There is no marine drug. The study is out of scope.

Abstract: The PEG term appears suddenly with no relation to the prior text.

A single line cannot be a paragraph.

The introduction provides some knowledge about the materials. It fails to hypothesize the goal. It does not constitute prior arts that are required to build project novelty. The study is conventional.

low molecular weight chitosan (50 kDa with a deacetylation degree ≤ 25%- 121 Sigma-Aldrich Ref. 448869) – such a low degree of deacetylation is abnormal.

Section 2.1: Indicate functions of excipients.

Is albumin positively charged under the processing pH?

The naming is confusing. Do nanoparticles contain PEG, poloxamer or insulin? Check text throughout for consistent presentation.

Section 2.2.4: Do you assay dissolution medium? If not, you should test remaining insulin content in nanoparticles, not just supernatant.

In results, without assaying chitosan, albumin and PEG content in coat, an accurate mechanistic insight cannot be provided. In fact, PEG may not be able to stay in coat. The present manuscript is only reporting and can only report physical outcomes – a screening study.

Author Response

Responses to Reviewer #3

The same nanoparticle design has been published in J Control Release, 2016 Jun 28;232:29-41. doi: 10.1016/j.jconrel.2016.04.012. Epub 2016 Apr 10. Dual chitosan/albumin-coated alginate/dextran sulfate nanoparticles for enhanced oral delivery of insulin.

R: The authors acknowledge and take the reviewer’s comments very seriously. However, the cited publication refers to alginate/dextran sulfate cores prepared by emulsification/internal gelation, whereas the nanoparticles described in this manuscript were prepared by ionotropic gelation followed by complexation.

There is no marine drug. The study is out of scope.

R: Authors are very concerned about biomaterials source as it can easily be checked through our publishing record. Although we cannot change insulin's origin as it is biotechnological, most of the polymers are marine provenience, thus fitting well on this journal.

Abstract: The PEG term appears suddenly with no relation to the prior text.

R: The authors appreciate the reviewer’s comments, and changes made to the abstract can be found as red text on page 1, line 13 of the revised manuscript.

A single line cannot be a paragraph.

R: The authors could not follow the Reviewer's point.

The introduction provides some knowledge about the materials. It fails to hypothesize the goal. It does not constitute prior arts that are required to build project novelty. The study is conventional.

R: Authors respect reviewer comments, but they do disagree. There is prior art pointing out a problem to solve, which is the uncertainty of the alginate/dextran sulfate core coating effect on nanoparticle physicochemical properties and insulin release in simulated GI media. No previous studies have disclosed the effect of dual coating of those cores with chitosan and albumin using a QbD approach while keeping nanoparticles' physicochemical properties during characterization. Only a strong knowledge of biopolymers and process parameters could make possible the identification of critical factors such as albumin, chitosan and PEG concentration. This optimization design was used to model the curvature in the relationship between the independent, at least three, and dependent factors and allowed us to find values of our factors to minimize or maximize a response or to hit a specific target.

low molecular weight chitosan (50 kDa with a deacetylation degree ≤ 25%- 121 Sigma-Aldrich Ref. 448869) – such a low degree of deacetylation is abnormal.

R: Actually, the degree of acetylation is ≤ 25%. Authors have corrected accordingly as it can be found on page 3, line 140 of the revised manuscript.         

Section 2.1: Indicate functions of excipients.

R: Alginate and dextran sulfate are polyanionic polymers that interact electrostatically with positively charged insulin and polymers under experimental pH conditions. Chitosan is a polycation under experimental conditions, poloxamer is a surfactant, PEG is a nanoparticle stabilizer, and calcium triggers alginate gelation.

Is albumin positively charged under the processing pH?

R: Albumin is predominantly positively charged under the processing pH.

The naming is confusing. Do nanoparticles contain PEG, poloxamer or insulin? Check text throughout for consistent presentation.

R: Nanoparticles contain insulin, PEG and poloxamer. However, poloxamer was not the focus of this experimental work as the focus was the influence of PEG, chitosan and albumin on dependent variables.  The text has been read carefully throughout.

Section 2.2.4: Do you assay dissolution medium? If not, you should test remaining insulin content in nanoparticles, not just supernatant.

R: The authors understand the reviewer’s concern about the remaining insulin not being quantified during release studies. We assayed the dissolution medium, and no insulin was found. When albumin is present, it contaminates insulin quantification using the direct method. We had no choice than to use the indirect method. Even though it is a relevant topic, it falls out of the scope of the manuscript. Considering the Reviewer's comments, the authors improved part of the manuscript related to the results of release studies, which can be found in red text on page 9, lines 344-347 of the revised manuscript.

In results, without assaying chitosan, albumin and PEG content in coat, an accurate mechanistic insight cannot be provided. In fact, PEG may not be able to stay in coat. The present manuscript is only reporting and can only report physical outcomes – a screening study.

R: We have not quantified any of those polymers, so we do not have the remaining polymers. These nanoparticles have been studied, including a mechanism over the last decade. We believe the experimental work stands on its own, and “in vitro” and “in vivo” assays related to insulin pharmacological action will be undertaken soon. This design, rather than an initial experiment at the initial stages of experimentation, is an optimization design as these biopolymer-based nanoformulations have been tested for over a decade. Only a strong knowledge of biopolymers and process parameters could make possible the identification of critical factors such as albumin, chitosan and PEG concentration. This optimization design was used to model the curvature in the relationship between the independent, at least three, and dependent factors and allowed us to find values of our factors to minimize or maximize a response or to hit a specific target.

Reviewer 4 Report

The manuscript “Chitosan/Albumin Coating Factorial Optimization of Alginate/Dextran Sulfate Cores for Oral Delivery of Insulin” by Bruno Pessoa, Mar Collado Gonzalez, Giuseppina Sandri, and António J Ribeiro is devoted to optimizing a nanoparticle formulation for oral delivery of insulin by assessing the relationship between design parameters and experimental data using response surface methodology through a 3-factor 3-level optimization Box-Behnken method. The authors used various materials to form multilayer protection of insulin inside nanoparticles.

There is no doubt that the area of research is important and the manuscript is worth to be published. However, the reviewer expects to improve/modify the manuscript before publication. The most crucial points are as follows:

·         In the introduction, authors should consider recent works in the field of innovative insulin delivery, primarily by citing work on systems that are responsive to changes in blood sugar levels, being of clinical importance;

·         From the clinical perspective which should be considered as the final goal, the more important parameter than the global release of insulin is the kinetics of insulin release. Moreover, the authors should be aware, that such a system is a kind of passive (non-responding) system and its clinical significance is minimal if not none.  So, the factorial optimization applied by the authors should be considered important from the pure scientific perspective, only.

·         All experimental results should be provided with determination error

·         The last two sentences in Conclusions are not well supported by results and should be reformulated. They are also not clear. What the authors mean by writing in the last sentence “Compared to previous nanoparticle-based formulations, this PEG/chitosan-induced……….”

Author Response

Responses to Reviewer #4

Reviewer #4

The manuscript "Chitosan/Albumin Coating Factorial Optimization of Alginate/Dextran Sulfate Cores for Oral Delivery of Insulin" by Bruno Pessoa, Mar Collado Gonzalez, Giuseppina Sandri, and António J Ribeiro is devoted to optimizing a nanoparticle formulation for oral delivery of insulin by assessing the relationship between design parameters and experimental data using response surface methodology through a 3-factor 3-level optimization Box-Behnken method. The authors used various materials to form multilayer protection of insulin inside nanoparticles.

There is no doubt that the area of research is important and the manuscript is worth to be published. However, the reviewer expects to improve/modify the manuscript before publication. The most crucial points are as follows:

R: We would like to thank the reviewer for the valuable comments, which improved the overall quality of the manuscript. We addressed them to the best of our knowledge, and the answers to each comment can be found below.

  • In the introduction, authors should consider recent works in the field of innovative insulin delivery, primarily by citing work on systems that are responsive to changes in blood sugar levels, being of clinical importance;

R: The latest developments are associated with a formulation that allows for weekly insulin injection and a potential formulation for oral administration. The first concerns insulin icodec. The results of phase 2 studies were published in November 2020 and revealed that the treatment "was well-tolerated and had a glucose-lowering efficacy and a safety profile similar to those of once-daily insulin glargine U100 in patients with type 2 diabetes" [1]. More recently, on October 3, 2022, the phase 3a study "ONWARDS 5" revealed the superiority of insulin icodec in decreasing HbA1c compared to once-daily insulin in patients with type 2 diabetes [2]. The second concerns ORMD-0801, a patent formulation that includes composition protection against protease degradation in the gastrointestinal tract and absorption enhancers at the intestinal level. It is currently in phase 3 studies, but according to the press release published on January 12, 2023, on the clinical trial sponsor's website, the endpoints were not met [3]. Following the reviewer's suggestion, we included literature data in the introduction section of the main text. Please see section 1, page 1, lines 29-40.

  • From the clinical perspective which should be considered as the final goal, the more important parameter than the global release of insulin is the kinetics of insulin release. Moreover, the authors should be aware, that such a system is a kind of passive (non-responding) system and its clinical significance is minimal if not none.  So, the factorial optimization applied by the authors should be considered important from the pure scientific perspective, only.

R: The authors appreciate and understand the reviewer's comment. However, this work focuses on the multiparametric approach in the coating of insulin-loaded alginate/dextran sulfate cores to protect insulin stability and provide appropriate release of this peptidic drug in the gastrointestinal tract. The authors agree that a delivery system capable of mimicking basal insulin release would be of greater importance from a clinical perspective, and these nanoparticle delivery systems, rather than passive delivery systems, are pH responsive instead. Before pursuing preclinical and clinical assays, authors are focused on a complete and comprehensive characterization of nanoparticles which using the QbD concept is the primary goal of this manuscript.

  • All experimental results should be provided with determination error.

R: All experimental results were statistically analyzed according to the methodology described in section 2.2.9. Furthermore, the minimum significance level for comparison between samples was set at a probability of p < 0.05. Considering the reviewer’s comments, the authors created an extra column in table 3 to provide the lower and upper value of coefficients at 95% confidence interval for each regression coefficient, as found in red text on page 11.

  • The last two sentences in Conclusions are not well supported by results and should be reformulated. They are also not clear. What the authors mean by writing in the last sentence "Compared to previous nanoparticle-based formulations, this PEG/chitosan-induced………."

R: We appreciate the reviewer's comment. The sentence concerns formulations developed in previous work in [4] and [5] whose composition is similar but not the same. Following reviewer’s comments, the authors reviewed the conclusion section, and the last sentence was corrected to increase clarity, as it can be found in red text on page 14, lines 496-497.

References

[1]       Bajaj HS, Bergenstal RM, Christoffersen A, et al. Switching to Once-Weekly Insulin Icodec Versus Once-Daily Insulin Glargine U100 in Type 2 Diabetes Inadequately Controlled on Daily Basal Insulin: A Phase 2 Randomized Controlled Trial. Diabetes Care. 2021;44:1586–1594.

[2]       Once-weekly insulin icodec demonstrates superior reduction in HbA1c in combination with a dosing guide app versus once-daily basal insulin in people with type 2 diabetes in ONWARDS 5 phase 3a trial. https://www.novonordisk.com/news-and-media/news-and-ir-materials/news-details.html?id=138024. (February, 2023)

[3]       Oramed Announces Top-line Results from Phase 3 Trial of ORMD-0801 for the Treatment of Type 2 Diabetes - Oramed Pharmaceuticals. https://oramed.com/oramed-announces-top-line-results-from-phase-3-trial-of-ormd-0801-for-the-treatment-of-type-2-diabetes/. (February, 2023)

[4]       Collado-González M, Ferreri MC, Freitas AR, et al. Complex Polysaccharide-Based Nanocomposites for Oral Insulin Delivery. Mar Drugs. 2020;18:55.

[5]       Woitiski CB, Veiga F, Ribeiro A, et al. Design for optimization of nanoparticles integrating biomaterials for orally dosed insulin. Eur J Pharm Biopharm. 2009;73:25–33.

Round 2

Reviewer 1 Report

I agree with the reviewer #3 that the manuscript is basically a screening study. Although the author proved that the optimized properties of nanoparticle can be achieved using the Box-Behnken design, especially for the variable setting they selected, the true physiochemical mechanistic insight remained unknown. In addition, I doubted that PEG was still in the nanostructure. Last, the author claimed that the nanoparticle was optimized but no in vitro or in vivo evidence were provided to prove the statement. To consider to be published in Marine Drugs, I suggest the author to address the issue provided as follows:

1.      Define the “optimum nanoparticle”: what is the definition of the optimum nanoparticle in the manuscript? As in vitro and in vivo study was not performed in the manuscript, I suggest the author state the limitation and address why the nanoparticle is optimized? Please support the statement by citing reliable publication with clearly defined materials parameters window including size, zeta potential, pH sensitivity, release profile…etc. A table might be suitable to support the results for the manuscript.

2.      The difference in composition cannot be recognized as novelty in my point of view. The author should discuss more on their strategy on selecting the independent and dependent variables in Box-Behnken design, and what is the advantage of the selection compared with other works?

3.      Since the author claimed in the response that this is the first paper to discuss ionotropic gelation/complexation focused on the chitosan and albumin dual coating step. I would suggest the authors to provide more discussion on it and explain the advantage of the complexation between chitosan and albumin compared with prior work.

4.      The preparation of nanoparticles is a two-step process, which involved the formation of a pregel core and the further coating by polyelectrolyte complexation  with oppositely charged materials. Please provide the zeta potential of the pregel core to confirm the charge is favorable for further chitosan and albumin coating.

5.      As the author replied in the response, BSA presented a zeta potential close to 0 mV at pH 4.6. From the literature it has an isoelectric point between pH = 4.7 and 5. Therefore, we can see the zeta potential for BSA is somewhere close to the coating environment where the was pH= 4.6. As BSA was without positive-charged, how did it facilitate the electrostatic complexation with the negatively-charged core? Please address the question in the manuscript. 

Author Response

I agree with the reviewer #3 that the manuscript is basically a screening study. Although the author proved that the optimized properties of nanoparticle can be achieved using the Box-Behnken design, especially for the variable setting they selected, the true physiochemical mechanistic insight remained unknown. In addition, I doubted that PEG was still in the nanostructure. Last, the author claimed that the nanoparticle was optimized but no in vitro or in vivo evidence were provided to prove the statement. To consider to be published in Marine Drugs, I suggest the author to address the issue provided as follows:

  1. Define the “optimum nanoparticle”: what is the definition of the optimum nanoparticle in the manuscript? As in vitro and in vivo study was not performed in the manuscript, I suggest the author state the limitation and address why the nanoparticle is optimized? Please support the statement by citing reliable publication with clearly defined materials parameters window including size, zeta potential, pH sensitivity, release profile…etc. A table might be suitable to support the results for the manuscript.

R: We acknowledge the time spent and efforts made by the reviewer to improve our manuscript. The definition of “optimum nanoparticle” toward desired physiochemical characterization is a nanoparticle with the following physicochemical properties:

Minimum particle size, minimum polydispersity index (PDI) and zeta potential less than -30 mV were assessed as described in the methodology, e.g., all of these attributes revealed stability over a one-week time. Among these formulations, we searched using the QbD concept for those protecting encapsulated insulin from the gastrointestinal system under gastric conditions and providing maximum bioactive insulin release under intestinal conditions. This nanoparticle optimization is valid for the evaluated variables including “in vitro” release study but validity does apply for variables not evaluated among which are mucoadhesion, intestinal uptake or antihyperglycemic properties. These are tasks to be performed in the near future using “in vitro” and “in vivo” assays” but before proceeding authors wanted to make sure that an optimized formulation regarding physicochemical properties could be found using the QbD concept

The discussion of results was improved upon a comparison of obtained results with similar previous nanoparticle-based formulations as it can be found in red text on page 4, lines 177-189. All the added cited publications are reliable as they are the remaining cited works. Authors made efforts to follow the reviewer’s point while comparing results using a table but easily found that different characterization methodologies are a major bottleneck toward objective and comparable results, i.e., a comprehensive characterization.

A statement about the validity and scope of this optimized nanoparticle formulation is provided in the Conclusions section of the manuscript as follows:

“Compared to previous nanoparticle-based formulations prepared using the same protocol, the QbD-driven optimized formulation resulting from this work showed a narrow size distribution induced by the PEG/chitosan ratio.”

This statement and others in the Conclusions are not beyond methodology, results and its discussion. Further changes in the Conclusion would result probably in redundancy therefore not adding value to the manuscript.

  1. The difference in composition cannot be recognized as novelty in my point of view. The author should discuss more on their strategy on selecting the independent and dependent variables in Box-Behnken design, and what is the advantage of the selection compared with other works?

R: Authors understand the reviewer’s point but the advantages of using Box-Behken design rather than of factor at the time (OFAT) can be found on page 2, lines 102-109. Taking into consideration the reviewer's comments, changes have been made in the discussion upon explanation of selected independent and dependent variables, as can be found in red text on page 3, lines 142-145.

  1. Since the author claimed in the response that this is the first paper to discuss ionotropic gelation/complexation focused on the chitosan and albumin dual coating step. I would suggest the authors to provide more discussion on it and explain the advantage of the complexation between chitosan and albumin compared with prior work.

R: The novelty of this study rather than the first approach to discuss ionotropic gelation/complexation focused on the chitosan and albumin dual coating step lies in the optimization of the chitosan/albumin coating step of alginate/dextran sulfate cores for oral delivery of insulin instead. Therefore, the authors cannot explain what this manuscript did not propose to.

  1. The preparation of nanoparticles is a two-step process, which involved the formation of a pregel core and the further coating by polyelectrolyte complexation  with oppositely charged materials. Please provide the zeta potential of the pregel core to confirm the charge is favorable for further chitosan and albumin coating.

R: The authors acknowledge the reviewer's comments. The alginate/dextran sulfate cores were coated first with chitosan, a predominantly positively charged polymer at the experimental pH, and then with albumin, a protein with charges balanced at the experimental pH. The zeta potential of the preformed cores using ionotropic gelation was -36 mV. The authors made changes in the manuscript as it can be found in red text on page 4. Lines 177-179.

Reviewer 2 Report

I recommended for publication after minor revision  (English language and style are fine/minor spell check required )

Author Response

We acknowledge the time spent and efforts made by the reviewer to improve our manuscript.

Reviewer 4 Report

The changes made by the authors are enough to accept the manuscript for publication

Author Response

(The authors gave the same response as above.)

Round 3

Reviewer 1 Report

1.         The term of “QbD-driven optimized formulation” based on the current description in the manuscript is not quite appropriate. First, quality by design (QbD) should be given in full name in its first appearance. Second, we did not see the full quality plan of the QbD strategy. According to ICH Q8(R2) guideline, QbD is “A systematic approach to development that begins with predefined objectives and emphasizes product and process understanding and Process control, based on sound science and Quality Risk Management”. The authors might have attempted to define some parts of the target product profile, but other important parts including manufacturing process, critical quality attribute, and validation of analytical methodologies to monitor those parameters were not even covered in the manuscript. Therefore, I would suggest the author to remove the term.

2.         Page 4 line 179, the author mentioned that the coating of chitosan did not reverse the zeta potential of the nanoparticle. They provided the readers with reference 43 to support the phenomenon; however, in contrast, in the reference, the zeta potential shifted from -16 mV to +15.1 mV after chitosan coating. In fact, in the same paper, similar materials and coating strategy (Albumin-chitosan-PEG-coated alginate-dextran-sulfate-core nanospheres) was performed to encapsulate and protect insulin.
The author should provide a table just like they did in Table 1 in reference 43 to justify the polyelectrolyte complexation process. 

Author Response

  1. The term of “QbD-driven optimized formulation” based on the current description in the manuscript is not quite appropriate. First, quality by design (QbD) should be given in full name in its first appearance. Second, we did not see the full quality plan of the QbD strategy. According to ICH Q8(R2) guideline, QbD is “A systematic approach to development that begins with predefined objectives and emphasizes product and process understanding and Process control, based on sound science and Quality Risk Management”. The authors might have attempted to define some parts of the target product profile, but other important parts including manufacturing process, critical quality attribute, and validation of analytical methodologies to monitor those parameters were not even covered in the manuscript. Therefore, I would suggest the author to remove the term.

R: We understand the concerns raised by the Reviewer regarding the terminology used by authors. We would like to add, however, that formulation was optimized using the Quality by Design (QbD) for factors disclosed in methodologies. For clarification purposes, rather than QbD-driven optimization, we used the terminology “factorial optimized” instead. Please, see page 11, line 390

  1. Page 4 line 179, the author mentioned that the coating of chitosan did not reverse the zeta potential of the nanoparticle. They provided the readers with reference 43 to support the phenomenon; however, in contrast, in the reference, the zeta potential shifted from -16 mV to +15.1 mV after chitosan coating. In fact, in the same paper, similar materials and coating strategy (Albumin-chitosan-PEG-coated alginate-dextran-sulfate-core nanospheres) was performed to encapsulate and protect insulin.
    The author should provide a table just like they did in Table 1 in reference 43 to justify the polyelectrolyte complexation process. 

R: Authors acknowledge the Reviewer's comments, which improved the quality of the manuscript. Experimental work supported by Reference 43 provided a different result, but a different amount of chitosan was used compared to this experimental work. Authors made changes in the revised manuscript toward clarity of this comparison between results obtained in these experimental works, as found in red text on page 11, lines 181-182. With regard to the suggestion to create a table to justify the polyelectrolyte complexation process, after thoughtful consideration, we have decided to maintain the discussion as it has been submitted as a detailed and comprehensive characterization of these nanoparticle-based formulations has been provided in a previous publication (Reference 13).